# Species-Specific Paternal Age Effects and Sperm Methylation Levels of Developmentally Important Genes

**DOI:** 10.3390/cells11040731

**Published:** 2022-02-19

**Authors:** Andreas Prell, Mustafa Orkun Sen, Ramya Potabattula, Laura Bernhardt, Marcus Dittrich, Thomas Hahn, Martin Schorsch, Federica Zacchini, Grazyna Ewa Ptak, Heiner Niemann, Thomas Haaf

**Affiliations:** 1Institute of Human Genetics, Julius Maximilians University, 97074 Würzburg, Germany; andreas.prell@stud-mail.uni-wuerzburg.de (A.P.); mustafa_orkun.sen@stud-mail.uni-wuerzburg.de (M.O.S.); ramya.potabattula@uni-wuerzburg.de (R.P.); laura.bernhardt@uni-wuerzburg.de (L.B.); marcus.dittrich@biozentrum.uni-wuerzburg.de (M.D.); 2Department of Bioinformatics, Julius Maximilians University, 97074 Würzburg, Germany; 3Fertility Center, 65189 Wiesbaden, Germany; th.hahn@mail.de (T.H.); martin.schorsch@gmx.de (M.S.); 4PERCUROS BV, 2333 CL Leiden, The Netherlands; f.zacchini@percuros.nl; 5Malopolska Centre of Biotechnology, Jagiellonian University, 30-387 Krakow, Poland; g.ptak@uj.edu.pl; 6Wolfson Centre for Age-Related Diseases, King’s College London, London SE1 1UL, UK; 7Clinic for Gastroenterology, Hepatology and Endocrinology, Medical University Hannover, 30625 Hannover, Germany; niemann.heiner@mh-hannover.de

**Keywords:** age-related differentially methylated regions (ageDMRs), bisulfite pyrosequencing, mammalian male germline, paternal age effect, species-specific epigenetic marks, sperm DNA methylation

## Abstract

A growing number of sperm methylome analyses have identified genomic loci that are susceptible to paternal age effects in a variety of mammalian species, including human, bovine, and mouse. However, there is little overlap between different data sets. Here, we studied whether or not paternal age effects on the sperm epigenome have been conserved in mammalian evolution and compared methylation patterns of orthologous regulatory regions (mainly gene promoters) containing both conserved and non-conserved CpG sites in 94 human, 36 bovine, and 94 mouse sperm samples, using bisulfite pyrosequencing. We discovered three (*NFKB2*, *RASGEF1C*, and *RPL6*) age-related differentially methylated regions (ageDMRs) in humans, four (*CHD7*, *HDAC11*, *PAK1*, and *PTK2B*) in bovines, and three (*Def6*, *Nrxn2*, and *Tbx19*) in mice. Remarkably, the identified sperm ageDMRs were all species-specific. Most ageDMRs were in genomic regions with medium methylation levels and large methylation variation. Orthologous regions in species not showing this age effect were either hypermethylated (>80%) or hypomethylated (<20%). In humans and mice, ageDMRs lost methylation, whereas bovine ageDMRs gained methylation with age. Our results are in line with the hypothesis that sperm ageDMRs are in regions under epigenomic evolution and may be part of an epigenetic mechanism(s) for lineage-specific environmental adaptations and provide a solid basis for studies on downstream effects in the genes analyzed here.

## 1. Introduction

The sperm epigenome is the end product of male germline reprogramming, which is affected by stochastic and environmental factors, including male infertility, paternal diet, and aging [1,2]. The methylation status of promoter elements and other regulatory regions is critically involved in shaping gene expression profiles during development and differentiation [3]. Accumulating evidence indicates that the sperm methylome affects the embryo development and disease susceptibility of the resulting offspring [4,5,6].

Historically, medical problems associated with delayed parenthood were primarily attributed to maternal aging. The decreasing ovarian reserve and increasing oocyte aneuploidy rate are associated with serious fertility problems, miscarriages, and children with Down syndrome [7]. However, the developmental potential of sperm from aging men is also reduced [8]. The increasing rate of de novo genetic mutations in the offspring of older males elevates the risks for some rare monogenic [9] and complex, in particular, neurodevelopmental disorders [10]. The number of spermatogonial cell divisions increases from 35 times at puberty to > 800 times at the age of 50 years [9]. During each replication cycle, not only the DNA sequence itself, but also epigenetic marks must be correctly copied to the daughter cells. Since the error rate of this copying process is estimated to be 10–100 times higher for epigenetic than for genetic information [11], the spermatozoa from older males have accumulated many more epimutations than DNA sequence mutations. In the aging mouse model, sperm DNA methylation changes have been associated with changes in gene methylation and expression in the brain and abnormal behavior in the offspring derived from older males [12].

The age-related gain in ribosomal DNA (rDNA) methylation reflects functional changes in nucleolar biology during aging and in age-related conditions [13,14]. Moreover, the correlation between rDNA methylation and aging has been conserved across a broad spectrum of somatic tissues and in the male germline in different mammalian species [2,15]. This evolutionary conservation is generally considered a good indicator of functional significance.

Paternal age effects have been extensively studied in the human [4,16,17,18], bovine [19,20,21], and mouse [12,22,23] sperm methylomes. Several epigenetic clocks, derived by linear regression algorithms on different methylation array data, have been successfully used for human sperm age prediction [16,17,18,24]. However, there is little overlap between the identified ageDMRs in different data sets and the highly selected CpGs (scattered throughout the genome) for different epigenetic clocks. The sperm epigenome has undergone extensive genome-wide methylation reprogramming in the male germline [25], which may explain the small intersection between age-related CpGs in sperm and somatic tissues such as blood [26,27].

In humans, CpGs susceptible to age-related sperm methylation changes appear to be enriched in the proximity of genes thought to be critically involved in embryogenesis and neuronal development, thereby supporting a role for the aging sperm methylome on reduced developmental potential and increased life-long disease risk of the offspring [4,17]. However, the relationship between the highly selected target CpGs of different epigenetic clocks and the aging process remains an enigma [26,27]. The goal of our study was to test whether the paternal age effect exists at the single-gene level of orthologous regulatory regions in humans, bovines, and mice.

## 2. Materials and Methods

### 2.1. Study Samples

The study on human (*Homo sapiens*, HSA) sperm samples was approved by the ethics committee at the medical faculty of the University of Würzburg (no. 117/11 and 212/15). After in vitro fertilization (IVF) or intracytoplasmic sperm injection (ICSI) at the Fertility Center Wiesbaden, the left-over swim-up sperm fraction (excess material) was collected, pseudonymized, and frozen at −80 °C until further use. To eliminate contamination by bacteria, lymphocytes, epithelial, and other somatic cells, the swim-up sperm samples were gently thawed and purified further by density gradients PureSperm 80 and 40 (Nidacon, Mölndal, Sweden). The vast majority (92 of 94) of sperm samples were from males with normal semen parameters.

Thirty-six sperm samples from 15 high-performance breeding bulls (*Bos taurus*, BTA) were obtained from Masterrind, Verden, Germany. Two or three samples (collected at young, middle, and old age) were available from 12 bulls. Bull sperm samples were purified by BoviPure and BoviDilute (Nidacon). Ninety-four mouse (*Mus musculus*, MMU) sperm samples were isolated from 3 to 16-month-old mice after cervical dislocation. The vas deferens and caudal epididymis were dissected and placed separately into 500 µL GMOPS with 10 mg/mL human serum albumin at 37 °C. Swim-up sperm purification was performed, and the final fraction was washed twice with PBS and resuspended in 500 µL 1× PBS.

For DNA isolation, the purified sperm cells were resuspended in 300 µL buffer (5 mL of 5 M NaCl, 5 mL of 1 M Tris-HCl; pH 8, 5 mL of 10% SDS; pH 7.2, 1 mL of 0.5 M EDTA; pH 8, 1 mL of 100% β-mercaptoethanol, and 33 mL of H_2_O), and 100 µL (20 mg/mL; 600 mAU/mL) proteinase K (Qiagen, Hilden, Germany), and incubated for 2 h at 56 °C. Sperm DNA was isolated using the DNeasy Blood and Tissue kit (Qiagen). DNA concentration and purity were measured by NanoDrop 2000c spectrophotometer (Thermo Scientific, MA, USA). Bisulfite conversion was carried out using the EpiTect Fast 96 Bisulfite kit (Qiagen) following the manufacturer’s recommendations. Bisulfite-converted DNA samples were stored at −20 °C until further use.

### 2.2. Study Genes

Candidate genes were selected from an in-house data bank, based on reduced representation bisulfite sequencing (RRBS) on 73 human, 16 bovine, and 24 mouse sperm samples (unpublished results). Three genes, *NFKB2*, *RASGEF1C*, and *RPL6*, were selected from a preliminary list of ageDMRs in human sperm. Similarly, four genes, *CHD7*, *HDAC11*, *PAK1*, and *PTK2B*, were selected from the bovine and three, *Def6*, *Nrxn2*, and *Tbx19*, from the mouse list (Appendix A). Orthologous regions in the human, bovine, and mouse ageDMRs as well as conserved CpG sites in the three studied species were identified with the Ensembl BLAST tool. Genome Reference Consortium Human Build 38 (GRCh38)/hg38, ARS-UCD1.2/bosTau9, and Genome Reference Consortium Mouse Build 38 (GRCm38)/mm10 were used as reference genomes. It is important to note that there is no information on the methylation levels of the identified sperm ageDMRs in other tissues or cell types.

### 2.3. Bisulfite Pyrosequencing

Polymerase chain reaction (PCR) and sequencing primers (Appendix A) for orthologous human, bovine, and mouse amplicons were designed using the Pyro-Mark Assay Design 2.0 software (Qiagen). DNA methylation standards with 0%, 50%, and 100% methylation were used for assay establishment. PCR for each sample was performed in a 25 µL reaction consisting of 2.5 µL 10×PCR buffer with MgCl_2_, 0.5 µL (10 mM) dNTPs, 1.25 µL (10 pmol/mL) of each reverse and forward primer, 0.2 µL (5 U/µL) FastStart Taq DNA polymerase (Roche Diagnostics, Mannheim, Germany), 1 µL (~25 ng) bisulfite-converted DNA and 18.3 µL dH_2_O. PCR amplifications were carried out with an initial denaturation at 95 °C for 5 min, 35 cycles of 95 °C for 30 s, primer-specific annealing temperature (Appendix A) for 30 s, and 72 °C for 45 s, and a final extension step at 72 °C for 10 min. Pyrosequencing was carried out using Pyro Q-CpG software (Qiagen) and PyroMark Gold Q96 CDT reagent kit on the PyroMark Q96 MD system. Unmethylated and fully methylated DNA standards (Qiagen) were used as controls in each pyrosequencing run.

### 2.4. Statistical Analysis

Statistical analysis was performed using IBM SPSS version 26. The donor age was correlated with the sperm DNA methylation level of the corresponding amplicon at the individual CpG and the regional level. For human samples, Pearson’s partial correlations were applied to adjust for possible confounding factors such as sperm concentration and donor body mass index. Depending on the data distribution, Spearman’s correlations were used for bovine and mouse samples. A *p* value of <0.05 was considered as statistically significant throughout the analyses. To compare age-related methylation changes in orthologous regions across species, methylation of a given sample was adjusted to the lifespan, which is largely different between mice (28 months), bulls (20 years), and humans (80 years).

## 3. Results

Using bisulfite pyrosequencing in 94 human, 36 bovine, and 94 mouse sperm samples, to screen candidate genes for paternal age effects, we identified a number of ageDMRs in the human (*NFKB2*, *RASGEF1C*, and *RPL6*), bovine (*CHD7*, *HDAC11*, *PAK1*, and *PTK2B*), and mouse (*Def6*, *Nrxn2*, and *Tbx19*) sperm epigenomes, respectively. The DMRs were in the promoter (7 of 10), promoter-flanking (1 of 10), or regulatory (1 of 10) regions (Appendix A). The encoded genes are involved in transcriptional regulation, signaling, and neurodevelopment. Because of the rather high CpG mutation rate, even in the absence of DNA methylation [28,29], the orthologous regions in humans, bovines, and mice contain both evolutionarily conserved and non-conserved CpGs (Table 1). The donor age ranged from 29 to 72 years (mean ± SD; 39.3 ± 5.9) in humans, from one to 12 years (4.8 ± 3.1) in bovines, and from 3 to 16 months (8.9 ± 3.7) in mice. For humans, body mass index ranged from 19 to 32 kg/m^2^ (25.6 ± 2.9) and sperm concentration was from 15 to 260 million/mL (84.5 ± 45.8).

Since, usually, the density of methylated CpGs, rather than individual CpGs in the promoter or a regulatory region, turns a gene “on” or “off” [30], we first compared the average methylation of all CpGs in a target region between species (Table 2). Average sperm methylation of orthologous regions (mainly gene promoters) was not conserved across species. For example, *NFKB2*, *RASGEF1C*, and *RPL6*, which were endowed with human ageDMRs, displayed medium sperm methylation (20–80%) in humans, whereas the orthologous regions in bovines and mice were either hypermethylated (>80%) or hypomethylated (<20%). Similar was true for genes with bovine ageDMRs (*CHD7*, *HDAC11*, *PAK1*, and *PTK2B*) and mouse ageDMRs (*Def6*, *Nrxn2*, and *Tbx19*), respectively.

Next, we determined the correlation between donor age and the average methylation of orthologous regions for the 10 study genes (Figure 1). For a better graphical representation of the species-specific age effects, donor age was adjusted to the percentage of lifespan. Significant or highly significant age effects were observed for *NFKB2*, *RASGEF1C*, and *RPL6* in human sperm samples; for *CHD7*, *HDAC11*, *PAK1*, and *PTK2B* in bovine sperm samples; and for *Def6*, *Nrxn2*, and *Tbx19* in mouse sperm samples (Table 3). Strikingly, none of the 10 sperm ageDMRs were evolutionarily conserved. Either there was no significant correlation with donor age in the two other analyzed species (for example, see human *RPL6*) or we found a significant effect in the opposite direction (for example, see mouse *Def6* and *Tbx19*).

Finally, we performed correlation analyses of individual conserved CpGs (Appendix A). With one notable exception, results were identical to the regional methylation analysis. At the regional level, *Nrxn2* sperm methylation showed a significant negative correlation with donor age in mice, whereas no significant age effects were observed in humans and bovines (Figure 1; Table 3). The evolutionarily conserved *NRXN2* CpG in MMU19:6,504,094; HSA11:64,636,329, and BTA29:42,885,109 showed a highly significant (*p* < 0.001) loss of methylation in mice and humans, but not in bovines. Age-related methylation changes in the remaining 29 analyzed evolutionarily conserved CpG sites (in 10 genes) were all species-specific.

Although we cannot exclude the possibility of a selection bias, it is conspicuous that human and mouse sperm DMRs were negatively correlated with age, whereas bovine DMRs gained methylation with age (Figure 1, Table 3). Moreover, ageDMRs usually displayed medium methylation levels associated with considerable methylation variation. Hypermethylated and hypomethylated regions showed less methylation variation and appeared to be more resistant to paternal age effects.

## 4. Discussion

The main finding of this study is that sperm methylation patterns found in gene-regulatory regions that are susceptible to paternal age effects are largely species-specific. One can speculate that the species-specific paternal age effects on the sperm epigenome and their possible impact on gene regulation in the next generation [12] may be part of an evolutionarily conserved mechanism, to allow environmental adaptation in a species-specific manner. Overall, species differences in mammalian sperm epigenomes may be a driving force to shape lineage-specific complex phenotypes, e.g., brain functioning in humans, and lipid storage and metabolism in cattle [31].

Mammalian sperm show a bimodal distribution of hypermethylated and hypomethylated regions, with global methylation levels between 70% and 80% [31,32,33]. DNA methylation is important to prevent retrotransposon activity and genome instability in germ cells [34]. A seven-species comparison of mammalian sperm methylomes revealed that both the number and size of hypomethylated regions in sperm expanded during epigenome evolution [32]. Hypomethylated regions are frequently observed in regulatory elements, including promoters, enhancers, and insulators [35]. A fraction of methylation marks found on regulatory genetic elements, which escape genome-wide reprogramming in the germline [36] and after fertilization [37], are primary candidates for transgenerational epigenetic inheritance. The majority of hypomethylated sperm promoters are conserved across species [32] and may be involved in important biological processes in embryo development [31]. In contrast, evolutionarily conserved hypermethylated sperm promoters may be critically involved in the silencing of immune genes for adherence and implantation of the embryo in the uterine wall.

Since methylated, and to some extent also non-methylated, CpGs are mutational hotspots [28,29], the vast majority of CpG sites have diverged during mammalian evolution. Only about 400,000 CpG sites, mainly in coding regions, have been conserved across mammalian species [32]. Our results show that paternal age effects on sperm gene regulation are driven by both conserved and non-conserved CpGs in the target regions. This is not unexpected because the methylation of neighboring CpGs is highly interdependent and phenotypic effects through altered gene regulation are usually mediated by methylation changes at the regional level [30]. Only one of 30 studied evolutionarily conserved CpG sites was subject to paternal age effects (in the same direction) in two analyzed species (mouse and human), whereas the regional age effect was only observed in the mouse. It is tempting to speculate that the age-related loss of methylation at this particular CpG (MMU19:6,504,094) site in the mouse *Nrxn2* coding region served as a nucleation point for regional hypomethylation in the mouse. The lineage-specific formation of new hypomethylated regions, and an extension of existing hypomethylated regions, has been observed in mammalian sperm epigenome evolution [32].

Consistent with earlier comparisons of human, bovine, and mouse sperm methylomes [31], we also observed dramatic interspecies differences in the methylation levels of orthologous regions. In species showing a significant age effect, the methylation level of a given regulatory region was usually in the medium range (20–80%), and there was considerable methylation variation between individuals. The orthologous regions of the other two analyzed species were either hypermethylated or, less frequently, hypomethylated. Hyper- and hypomethylated regions exhibited less methylation variation and, rarely, a significant age effect (i.e., bovine *HDAC11*).

Although only a limited number of regulatory genes were analyzed, it is striking that selected human ageDMRs (*NFKB2*, *RASGEF1C*, and *RPL6*) lose methylation, whereas bovine DMRs (*RPL6*, *CHD7*, *HDAC11*, *PAK1*, *PTK2B*, *DEF6*, and *TBX19*) gain methylation with age. This is consistent with earlier studies on paternal aging effects in humans [4] and bovines [20]. In mouse sperm, three DMRs (*Def6*, *Nrxn2*, and *Tbx19*) lost, and one hypomethylated DMR (*Nfkb2*) gained methylation with age. One region (*RPL6*) showed opposite methylation changes in humans and bovines, one region (*NFKB2*) between humans and mice, and two regions (*DEF6* and *TBX19*) between bovines and mice. Provided that these age-related methylation changes are transmitted to the next generation, increased paternal age may be associated with gene activation in humans and mice, and gene silencing in bovines.

## 5. Conclusions

Although it is difficult to extrapolate the findings from candidate gene studies to the entire (epi)genome, our results reveal that, in contrast to rDNA and other repetitive elements [2], the paternal age effects on sperm methylation of individual genes are largely species-specific. Neither average methylation nor age-related methylation changes in orthologous regulatory regions are evolutionarily conserved. We hypothesize that sperm methylation and the regulation of individual genes in the resulting embryo may be part of an evolutionary mechanism to allow environmental adaptations in a species-specific manner.

Our study is restricted to gene-regulatory regions which are conserved between species. Age-related methylation changes involve neighboring conserved and non-conserved CpG sites. One advantage of our study is that bisulfite pyrosequencing is a highly accurate method (compared to genome-wide methylation screens) to quantify methylation at both the single-CpG and the regional level. The methylation differences between technical replicates (including bisulfite conversion) of our assays were in the order of 1–2 percentage points. Our assays allow one to analyze a relatively high number of sperm samples at reasonable cost and expenditure of time. This made it possible to detect even minor, but significant, age effects in hypomethylated (i.e., mouse *Nfkb2*) and hypermethylated (bovine *RPL6*, *DEF6*, and *TBX19*) regions with low methylation variation, which may escape detection in genome-wide screens. However, genes with medium methylation levels and large methylation variations appear to be more susceptible to paternal aging. This should allow the prioritization of candidate genes from genome-wide methylome data sets for future studies on sperm aging.

The male germline-specific reprogrammed sperm epigenome [25] is fundamentally different from the epigenomes of somatic cells. Although direct experimental evidence is lacking, it is unlikely that the identified sperm ageDMRs also exist in somatic tissues. The observed age-related sperm methylation changes in single copy genes were in the order of several percentage points and were comparable to those in previous studies [1,6]. Because of the enormous variation among individuals, there was considerable overlap in the methylation levels of a given gene in a given species between older and younger sperm donors. Small effect size does not necessarily exclude the functional importance of sperm methylation marks. Consistent with a multifactorial model, multiple age-related epigenetic changes may contribute to the developmental competence of the resulting embryos and the shaping of the health and disease of the offspring. Future studies on methylation and expression of the identified genes in embryonic, fetal, and adult tissues from the offspring of old vs. young fathers may directly demonstrate an impact of sperm age DMRs on the next generation.

## Figures and Tables

**Figure 1 cells-11-00731-f001:**
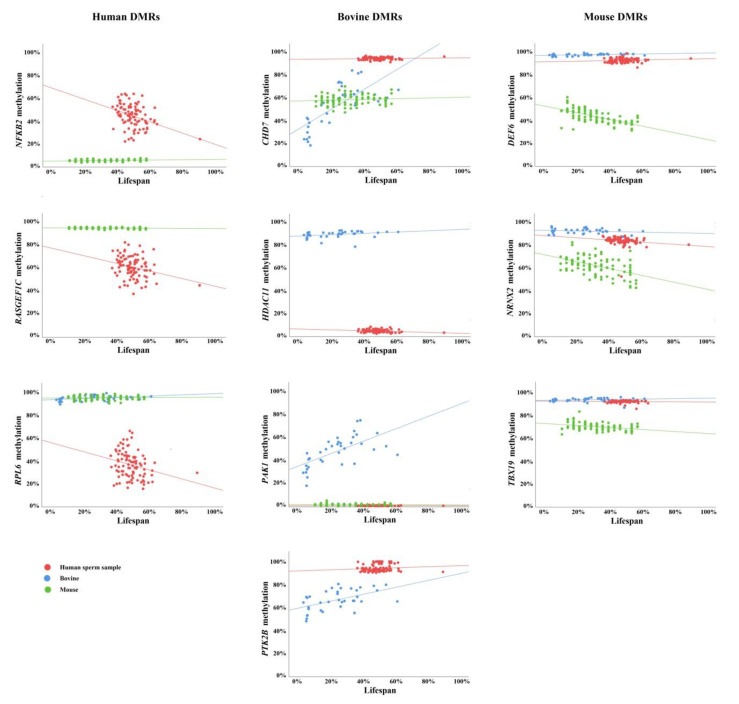
Scatter plots showing the correlations between average regional methylation, including conserved and non-conserved CpGs (y-axis in %) and donor age (x-axis in percentage of lifespan) in 94 human sperm samples (indicated by red dots), 36 bovine samples (blue dots), and 94 mouse samples (green dots). *NFKB2*, *RASGEF1C*, and *RPL6* are endowed with human-specific sperm ageDMRs; *CHD7*, *HDAC11*, *PAK1*, and *PTK2B* with bovine-specific sperm ageDMRs; and *Def6*, *Nrxn2*, and *Tbx19* with mouse-specific sperm ageDMRs.

**Table 1 cells-11-00731-t001:** Genes, genomic regions, and CpG sites that have been analyzed in human (HSA), bovine (BTA), and mouse (MMU) sperm.

Gene	Species	Genomic Localization ^a^	Sequence Analyzed by Bisulfite Pyrosequencing ^b^
**Human ageDMRs**			
*NFKB2*	HSA MMU	chr10: 102,398,797–102,398,849 chr19: 46,308,687–46,308,739	CGGGGGTGGCTCCCACATGGGTGGAGGCTCTGGGGGTGCAGCCGGGGGCTACG CGGGGGCGGATCCCACATGGGTGGAGGTTCTGGGGGCTCCGCTGGGGGTTATG
*RASGEF1C*	HSA MMU	chr5: 180,128,402–180,128,452 chr11: 49,960,606–49,960,556	CGGCTTACCAGTTCCACGTGGGTCAGCTGCTGGGCCAGTGTGTAGGGGTCG CCACTCACCAGCTCCACGTGGGTCAGTTGCTGGGCCAGCGTGTAGGGGTCA
*RPL6*	HSA BTA MMU	chr12: 112,408,273–11,240,8247 chr17: 61,838,759–61,838,785 chr5: 121,205,822–121,2058,48	CGGCGGTACCCGGGTGGTTAAACTTCG TGGTGGTACCCGAGTGGTCAAACTTCG CGGTGGCACCCGGGTGGTGAAGCTTCG
**Bovine ageDMRs**			
*CHD7*	BTA	chr14: 26,361,243–26,361,288	CTAGGCGGTTACCTGGCCCGGGGGGACTTCTCCATGCCGCAGCATG
	HSA	chr8: 60,741,997–60,742,042	ATGGGCAGCTATATGGCACGTGGGGATTTTTCCATGCAGCAGCATG
	MMU	chr4: 8,752,051–8,752,096	ATGGGCAGCTATCTGGCACGTGGGGATTTCTCCATGCAGCAGCACG
*HDAC11*	BTA HSA	chr22: 58,440,641–58,440,716 chr3: 13,481,361–13,481,287	CGGCGTC[…]AGCGCGGCGAGTACACGATGGGCCAGCG CGGCATC[…]AGCGCGGCGAGTACACGATTGGCCAGCG
*PAK1*	BTA HSA MMU	chr29: 18,586,633–18,586,667 chr11: 77,411,828–77,411,794 chr7: 97,843,083–97,843,117	CGGCTCTGCGACAGAAACCGCAGGCAGAGATGCCG CGGCTCTGCGACGGAAACAATCGCCAGAGATGCCG CGGCTCTGCGACAGATACACAAGATCATCAGAGAT
*PTK2B*	BTA	chr8: 74,491,018–74,490,993	CGACGTAATGTGCCCACCTTCACTCG
	HSA	chr8: 27,397,637–27,397,612	CGGCGTAACGTGCCCAACTTTACTCG
**Mouse ageDMRs**			
*Def6*	MMU HSA BTA	chr17: 28,217,012–28,217,055 chr6: 35,309,717–35,309,760 chr23: 9,282,363–9,282,406	CGTGGCCCTGGAGGAGCACTTCCGGGATGACGATGATGGCCCGG CGTGGCCCTGGAGGAACACTTCCGAGATGATGATGACGGCCCTG CGTGGCCCTGGAGGAGCACTTCCGAGACGATGACGATGGTCCCG
*Nrxn2*	MMU HSA BTA	chr19: 6,504,094–6,504,133 chr11: 64,636,291–64,636,330 chr29: 42,885,070–42,885,110	CGTTAGGGACCTAACA-CCCGCCCCCGGCAGCCGGATGGCG CGTTAGGGACCTCACA-CCCGCCCCCAGCAGCCGGCTGGCG CGTTAGGGACCTCACACCCCGCCCCCGGCAGCCGGATGGCA
*Tbx19*	MMU	chr1: 165,153,651–165,153,599	CGTCCGGACCG- ACAGTCACCGCTGGAAGTACGTCAATGGTGAATGGGTCCCCG
	HSA	chr1: 168,291,235–168,291,287	TGTCCCTA- CGGACAGTCACCGCTGGAAGTACGTCAACGGGGAATGGGTGCCCG
	BTA	chr3: 253,101–253,049	TGTCCCAA- CGGACAGTCATCGCTGGAAGTACGTCAATGGAGAATGGGTGCCTG

^a^ Genome Reference Consortium Human Build 38 (GRCh38)/hg38, ARS-UCD1.2/bosTau9, and Genome Reference Consortium Mouse Build 38 (GRCm38)/mm10 were used as references. ^b^ CpG sites which are conserved in at least two analyzed species are highlighted by different colors and non-conserved CpGs are shaded in gray.

**Table 2 cells-11-00731-t002:** Mean methylation of analyzed genomic regions in human, mouse, and bovine sperm.

Gene	Methylation (%) ± Standard Deviation (%) [Range (%)]		
	HSA (*n* = 94)	BTA (*n* = 36)	MMU (*n* = 94)
**Human ageDMRs**			
*NFKB2*	44.5 ± 10.0 [22.3–64.0]	n.d.	5.6 ± 0.8 [3.8–7.6]
*RASGEF1C*	60.1 ± 9.1 [37.0–81.2]	n.d.	93.5 ± 0.5 [91.8–94.6]
*RPL6*	36.8 ± 11.5 [15.8–66.5]	95.1 ± 2.2 [89.8–99.6]	95.6 ± 1.8 [90.7–99.0]
**Bovine ageDMRs**			
*CHD7*	93.7 ± 1.0 [90.6–96.2]	53.0 ± 18.1 [18.2–83.4]	57.9 ± 4.2 [46.9–69.5]
*HDAC11*	4.7 ± 1.2 [2.5–8.5]	89.3 ± 3.0 [78.6–92.9]	n.d.
*PAK1*	1.0 ± 0.1 [0.6–1.7]	48.1 ± 13.1 [18.3–75.0]	1.9 ± 0.6 [0.5–5.4]
*PTK2B*	94.0 ± 3.0 [90.3–100]	66.6 ± 8.7 [48.1–80.4]	n.d.
**Mouse ageDMRs**			
*Def6*	93.1 ± 1.9 [86.8–99.0]	97.9 ± 0.8 [96.0–99.7]	43.5 ± 5.9 [31.4–60.7]
*Nrxn2*	82.6 ± 3.8 [51.8–87.0]	91.1 ± 2.1 [86.3–95.1]	61.3 ± 8.1 [42.0–81.3]
*Tbx19*	93.3 ± 1.0 [86.9–94.7]	94.8 ± 1.6 [88.0–97.2]	71.1 ± 3.2 [64.2–84.4]

**Table 3 cells-11-00731-t003:** Correlations between donor age and mean methylation in human, bovine, and mouse sperm.

Gene	HSA (*n* = 94)		BTA (*n* = 36)		MMU (*n* = 94)	
	Pearson’s *r*	*p*	Spearman’s *ρ*	*p*	Spearman’s *ρ*	*p*
**Human ageDMRs**						
*NFKB2*	−0.37	0.002	n.d.		0.22	0.04
*RASGEF1C*	−0.28	0.03	n.d		-0.10	0.33
*RPL6*	−0.25	0.05	0.37	0.03	0.07	0.53
**Bovine ageDMRs**						
*CHD7*	0.09	0.47	0.75	< 0.0001	0.11	0.31
*HDAC11*	−0.22	0.08	0.50	0.002	n.d.	
*PAK1*	−0.01	0.92	0.68	< 0.0001	−0.06	0.56
*PTK2B*	0.11	0.42	0.53	0.001	n.d.	
**Mouse ageDMRs**						
*Def6*	0.10	0.42	0.40	0.02	−0.69	<0.0001
*Nrxn2*	−0.18	0.16	−0.15	0.37	−0.45	<0.0001
*Tbx19*	−0.05	0.71	0.49	0.003	−0.33	0.001

## Data Availability

The data underlying this work are available in the article and its online Appendix A.

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
