# Peer review of "Species-Specific Paternal Age Effects and Sperm Methylation Levels of Developmentally Important Genes"

_cells, 2022, doi:10.3390/cells11040731_

Round 1

Reviewer 1 Report

This is an interesting research focused on the effect of paternal age on the sperm methylation in different mammalian species.

The research is well-conducted and the results obtained are clearly presented and discussed.

Author Response

We are grateful for the overall very positive evaluation of our paper. There were no specific comments on our paper.

Reviewer 2 Report

This is a very interesting research conducted by the authors. However the authors have not been able to show here the enough range of experiments to be considered for this journal. The authors have performed the bisulfite sequencing in the entire study, and have drawn their final conclusion on the single set of experimentation. Differential methylation should be correlated with the expression of the genes, and other terminal affects on the cellular functions.

The authors should perform downstream experiments to report this beautiful study. 

Thank you

Author Response

The goal of our study was to test whether or not the paternal age effects on orthologous regulatory regions of single copy genes have been conserved during mammalian evolution. Our candidate gene analysis provides solid evidence that in contrast to repetitive rDNA elements, paternal age affects sperm gene methylation patterns in a species-specific manner. This is consistent with the idea that sperm age DMRs are subject to epigenomic evolution and may be involved in species-specific environmental adaptation. This study reports the first-ever high-resolution comparison of sperm ageDMRs in three different species and is based on a very substantial amount of work.

While being very positive on our paper, this reviewer suggests to add more data on the “downstream” effects of at least some of the genes analysed here. We feel this is far beyond the scope of the present study and we are currently planning to study the effects of age-related sperm methylation on gene expression in a follow-up study.

Reviewer 3 Report

This manuscript briefly depicts species-specific paternal age effects and sperm methylation levels of developmentally important genes. The manuscript is suitable for publication in the cells.

This article is well written. It provides in-depth discussion among sperm methylation levels of developmentally important genes in paternal age effects in a variety of mammalian species, including human, bovine, and mouse. The information obtained from whether or not paternal age effects on the sperm epigenome have been conserved in mammalian evolution and compared methylation patterns of orthologous regulatory regions containing both conserved and non-conserved CpG sites in human, bovine, and mouse sperm samples by bisulfite pyrosequencing. However, some of sentences are redundant, it has to be more concise and accurately reflect the points discussed.

Author Response

We would like to thank the reviewer for the comments and for the positive evaluation of our manuscript. As suggested, we have streamlined the discussion to avoid reiteration and be more concise.